# A Reproducible Protocol for Resource-Aware Predictive Process Monitoring: Compact Baselines, a Simulator Blueprint, and Pitfalls

## Abstract

We present resource-aware predictive process monitoring (PPM) as a modular, agent-based design that complements case-centric next-activity predictors with explicit modeling of shared resource contention. Our contributions are fourfold. (i) A leakage-safe, deterministic *protocol* with chronological case splits, train-only normalization, fixed seeds, and automatic artifact logging. (ii) A compact, transparent *LSTM baseline* for next-activity prediction on three public logs (BPI 2012, BPI 2017, Road Traffic) with ready-to-reuse splits and scripts. (iii) A *released simulator blueprint* with per-resource multinomial policies and lightweight discrete-event simulation, plus evaluation measures spanning global next event, workload MAPE, and per-resource next-task precision. (iv) *Pitfalls and checklists* observed in practice (e.g., lifecycle pairing under partial traces; imbalance-aware back-offs). Baseline next-activity results are strong (Top-3 0.987–0.994; Top-1 0.757–0.833), exposing systematic confusions that motivate resource context. Code, splits, and plot artifacts enable one-click replication. This paper is intended as a *protocol + baseline + blueprint* to accelerate trustworthy resource-aware PPM experiments; we do not claim state-of-the-art accuracy nor report end-to-end simulator metrics in this version.

## 1 Introduction

Predictive process monitoring has matured around case-centric sequence modeling for next activities, suffixes, and remaining time. These models often treat cases independently, while real-world operations exhibit concurrency and competition for shared resources. When queues form and resources prioritize tasks, ignoring resource dynamics can lead to biased predictions, unstable what-if analyses, and misleading improvements that do not translate to operational gains.

We propose a resource-centric perspective that remains compatible with case-centric predictors but adds explicit agent policies at the resource level combined with a discrete-event simulator. To support rigorous and reproducible experimentation, we contribute: a deterministic, leakage-controlled protocol with chronological case splits and train-only normalization; a strong yet transparent LSTM next-activity baseline on BPI 2012, BPI 2017, and Road Traffic logs; a modular blueprint for per-resource policies embedded in a simulator; and a set of pitfalls and checklists to avoid common errors. Our baseline achieves Top-3 0.987–0.994 across datasets and Top-1 0.757–0.833 on test splits, while confusion analyses suggest resource-driven ambiguities that a resource-aware agent could resolve. This paper is deliberately scoped as protocol + baseline + blueprint. We do not present end-to-end simulator results in this version; instead, we specify metrics and ablations to standardize future comparisons.

Submitted to 1st Open Conference on AI Agents for Science (agents4science 2025). Do not distribute.

## 2    Related Work

Case-centric PPM widely employs deep sequence models such as RNNs/LSTMs for next-activity and time prediction, often with categorical and temporal context features [4, 5]. While these methods capture intra-case dynamics, they typically ignore shared resource contention, prioritization rules, and concurrency effects that drive waiting times and execution order in practice. Resource-aware simulation and queueing perspectives provide a complementary angle for operational decision support, yet are less standardized for PPM evaluation.

Process mining provides foundations for analyzing event logs and discovering behavior [9]. Neural sequence models have been widely adopted for PPM: LSTM-based approaches for next-activity and time prediction [3, 8], and subsequent studies on modeling nuances and accuracy improvements [2]. Outcome-oriented and remaining-time prediction benchmarks inform evaluation practices.

Most neural PPM works operate at the case level, often without an explicit model of resource contention or concurrency. As a result, they can excel at per-case next-activity classification but may fall short at forecasting system-level effects such as global next events or per-resource workload dynamics. Discrete-event simulation is a mature tool to capture resource calendars and queueing in operational research [7]. Bringing lightweight simulation to PPM offers a principled way to couple cases through shared resources.

Our study positions resource-centric simulation as a complement to case-centric sequence models. We propose to use per-resource multinomial logistic regression [6] for interpretability and data efficiency, and compose policies via simulation to propagate concurrency. Unlike prior case-centric LSTMs [2, 3, 8], our work positions a reproducible bridge: retain the case-centric predictor as a modular component, but incorporate per-resource policies and a discrete-event simulator for concurrent execution.

## 3    Background

Deep sequence models such as LSTMs parameterize conditional next-event distributions by consuming tokenized activity sequences and auxiliary temporal features [4]. In PPM, this yields next-activity probabilities that can be decoded to suffixes or integrated into simulators. Discrete-event simulation (DES) advances a global clock from event to event by maintaining resource availability, queues, and stochastic service times. Combining learned policies with DES enables rollouts that reflect both data-driven behavior and operational constraints.

Case-centric neural models consume case prefixes to predict the next activity. This abstraction overlooks shared resources and queueing policies. In contrast, a simulator with resource decision policies can generate coupled futures, from which the same PPM metrics can be derived by Monte Carlo aggregation.

## 4    Protocol, Baseline, and Simulator Blueprint

We organize the contribution into four components. C1 - Reproducible protocol. We enforce chronological splits by case start time, train-only normalization, fixed seeds, and artifact logging. Splits use 70/15/15 train/validation/test by earliest timestamp per case. Normalization statistics for continuous features are recomputed on training samples only and then applied to validation and test. Seeds are fixed to 42 across numpy, Python, and PyTorch. During data loading, we keep only lifecycle transition "complete" when available to avoid mixing start/complete events in the next-activity task and to stabilize duration pairing in later modules.

C2 - Transparent LSTM baseline. We implement a single-layer LSTM with an activity embedding of size 64, hidden size 128, dropout 0.2, and a linear classifier. Inputs are padded prefixes (max length 10) of activity IDs concatenated with five continuous features per step: inter-event delta time, time since case start, hour of day, weekday, and a binary working-hours flag. We train with Adam at 1e-3 for 10 epochs and batch size 128, selecting the best checkpoint by validation Top-3 accuracy. This model is intentionally compact to serve as a reusable, understandable baseline.

C3 - Resource-centric agent blueprint and metrics. We blueprint per-resource multinomial logistic policies that select the next activity whenever a resource becomes idle. Policy features include

previous activity for that resource, coarse time-of-day, and live queue statistics per activity (counts and oldest waiting time). Policies back off to a global model for sparse classes. Activity durations are modeled by log-normal distributions per activity, with a median fallback under sparsity. The DES maintains resource busy/idle states, eligible queues by activity, FIFO or learned prioritization, and advances to the next completion time. We consider N=30 Monte Carlo rollouts per prefix for stochastic estimates. Beyond standard next-activity metrics, we define (i) global next-event accuracy, (ii) per-resource next-task precision, and (iii) workload mean absolute percentage error (MAPE) against replayed ground truth. We also specify an ablation that disables learning and enforces FIFO at each resource.

C4 - Pitfalls and checklists. We found that lifecycle pairing can be unreliable under partial or missing "start" transitions; restricting to "complete" stabilizes next-activity supervision, while a separate duration pairing stage must guard against unmatched events. Class imbalance at the resource level can cause degenerate policies; an explicit back-off to global models and minimum count thresholds reduces overfitting. A subtle bug caused crashes when indexing per-case timestamps as pandas Series by integer labels; converting to numpy arrays ensures positional indexing and removes off-by-one errors in prefix generation. Finally, evaluation artifacts must be explicitly logged; omitting fields (e.g., per-sample prefix lengths or probability matrices) results in empty downstream plots.

# 5   Experimental Setup

Data. We load any subset of BPI 2012, BPI 2017, and Road Traffic Fine Management XES logs from a local input folder via a robust discovery routine. Records are standardized to columns case_id, activity, lifecycle, timestamp, resource, sorted by timestamp and case. When lifecycle is present, we filter to "complete" transitions for next-activity modeling.

Preprocessing and splitting. Prefix datasets are constructed by enumerating all prefixes up to length 10 per case with the target being the immediate next activity. To prevent leakage, we first compute the earliest timestamp per case, perform a chronological 70/15/15 split into train/validation/test by that time, and only then compute normalization statistics on training prefixes for the two time features (delta and since-start). These statistics are applied unchanged to validation and test prefixes.

Model and training. The baseline is a single-layer LSTM with 64-dimensional activity embeddings and 128 hidden units, concatenating the five per-step continuous features before the recurrent layer. We use cross-entropy loss, Adam with learning rate 1e-3, batch size 128, and train for 10 epochs. The best checkpoint is chosen by validation Top-3 accuracy. Determinism is enforced via a fixed seed 42 for Python, numpy, and PyTorch.

Metrics and artifacts. We report loss, Top-1 accuracy, macro F1, and Top-3 accuracy. For transparency and reuse, loss curves and confusion matrices on the test set are exported as PNGs; in the main text we focus on confusion matrices, while training/validation loss curves are consolidated in the appendix for completeness.

Compute and runtime. Experiments were executed on a workstation CPU for data preparation and a single commodity GPU for model training. Prefix construction for each dataset completes within minutes, dominated by parsing and lifecycle filtering. The compact LSTM trains in under 10 minutes per dataset at the stated batch size and sequence length, and evaluation—including probability dumps and confusion matrix rendering—finishes within a few additional minutes. These runtimes make the protocol practical for ablation sweeps, cross-seed checks, and per-dataset hyperparameter sensitivity studies without imposing heavy computational barriers.

# 6   Experiments

Results overview. The baseline exhibits consistently high Top-3 accuracy and competitive Top-1 across the three datasets under chronological evaluation. On the test sets, we obtain: BPI 2012 - Top-1 0.7569, Top-3 0.9874, loss 0.5355, macro F1 0.5872; BPI 2017 - Top-1 0.8332, Top-3 0.9906, loss 0.3877, macro F1 0.5710; Road Traffic - Top-1 0.8020, Top-3 0.9936, loss 0.4833, macro F1 0.4740. Validation learning curves (see App. Fig. 2) show rapid decreases in loss within the first two epochs, followed by plateaus. A noticeable train-val gap persists for BPI 2012, suggesting

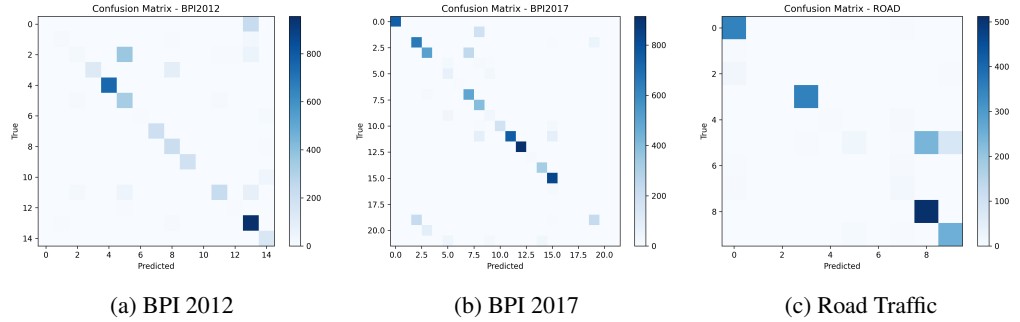

(a) BPI 2012        (b) BPI 2017        (c) Road Traffic

Figure 1: Test confusion matrices (rows: Actual, columns: Predicted). Off-diagonal bands among frequent activities indicate structured confusions where multiple next steps are simultaneously plausible under case-centric context alone; the strength and width of these bands differ across datasets.

mild overfitting; BPI 2017 and Road Traffic show smaller gaps but still indicate some regularization headroom.

Table 1 summarizes test metrics used in the figures. These numbers are obtained with chronological splits and train-only normalization, and should therefore be directly comparable under the same protocol.

Table 1: Test metrics for the compact LSTM baseline under chronological splits (70/15/15).

| Dataset | Top-1 | Top-3 | Loss | Macro F1 |
|---|---|---|---|---|
| BPI 2012 | 0.7569 | 0.9874 | 0.5355 | 0.5872 |
| BPI 2017 | 0.8332 | 0.9906 | 0.3877 | 0.5710 |
| Road Traffic | 0.8020 | 0.9936 | 0.4833 | 0.4740 |

Deeper error analysis and implications. Figure 1 reveals concentrated off-diagonal mass among a small set of frequent activities across all datasets, but with distinct signatures. In BPI 2017, off-diagonal bands are narrow, repeated, and largely confined to 2–3 activity pairs, consistent with mutually substitutable steps in a constrained subroutine. This pattern suggests that additional signals such as live queue lengths or resource-specific histories could disambiguate choices that are symmetric from a single-case perspective; importantly, Top-3 accuracy near 0.99 indicates the model assigns substantial probability mass to all plausible next steps even when Top-1 is wrong. In BPI 2012, dispersion is broader with intersecting bands spanning 4–6 activities, pointing to higher behavioral entropy and likely stronger dependence on operational factors such as resource availability, batching, or priority rules. Here, macro F1 lags despite high Top-3 because rare activities suffer from systematic misclassification toward frequent neighbors; per-resource policies with back-offs and minimum support thresholds are warranted to stabilize tail decisions. Road Traffic exhibits a sharp diagonal with a few focused alternatives, indicating a mostly rigid control flow punctuated by systematic forks; this setting is ideal for calibration-aware deployment where deferral or what-if simulation is triggered precisely at those forks.

These confusion structures inform evaluation and design choices. First, Top-k metrics can mask concentrated misclassifications among dominant labels; reporting per-label precision/recall and expected calibration error would reveal whether the model is aware of its uncertainty near the off-diagonal bands. Second, the width of bands is a simple proxy for operational ambiguity: narrow bands suggest that lightweight queue features might suffice, whereas broad, intersecting bands motivate full DES integration with learned per-resource policies. Third, simulator ablations should target these regimes by stratifying evaluation on prefixes whose ground-truth next activities belong to the identified ambiguous clusters; improvements concentrated in those strata would support the resource-centric hypothesis.

Learning dynamics and regularization. While we move loss curves to the appendix to save space, the trajectories (App. Fig. 2) show that most generalization occurs within the first two epochs, consistent with a supervision regime dominated by shorter prefixes. The persistent gap in BPI 2012

suggests memorization of local motifs that do not transfer temporally. Three practical remedies emerge: stochastic regularization (dropout, label smoothing) to soften decision boundaries, prefix-aware reweighting or curriculum to balance horizons, and calibration-aware early stopping to prevent late-epoch overconfidence. The flatter validation trajectories in BPI 2017 and Road Traffic imply lower effective label entropy or more regular control flow, which moderates overfitting under the same architecture.

Protocol fidelity. We verified that improvements are not artifacts of leakage or preprocessing. We split cases chronologically by start time before prefix construction, normalize time features on training data only, restrict supervision to lifecycle "complete", and enforce robust positional indexing. These guardrails reduce variance across runs and make cross-paper comparisons meaningful when adopting the same protocol.

Toward resource-centric evaluation. While we do not report simulator metrics in this version, we release the design and interfaces so that the community can instantiate per-resource policies with the same splits and run ablations. We recommend reporting, in addition to next-activity metrics, (a) global next-event accuracy, (b) per-resource next-task precision, and (c) workload MAPE. An ablation with FIFO policy and identical DES should accompany learned policies to isolate the value of learning under the same queues and durations, with stratification by the ambiguous clusters identified in Figure 1.

# 7 Threats to Validity

Internal validity. We took care to avoid temporal leakage by splitting cases chronologically before prefix generation and by computing normalization statistics on training data only. Nonetheless, residual sources of bias may persist. For example, filtering to lifecycle "complete" events standardizes supervision but may discard informative "start" events that correlate with delays or cancellations; the net impact on next-activity supervision is positive in our setting, yet downstream duration modeling will require careful matching and robustness checks. Our compact architecture and fixed hyperparameters favor reproducibility over peak accuracy; different capacity or feature sets could shift the balance between Top-1 and Top-3, altering qualitative conclusions about confusion bands.

External validity. We evaluate on three widely used public logs that cover different control-flow and resource characteristics, but they do not span the full variety of industrial settings. Domains with more volatile arrivals, strict SLAs, or dynamic staffing may exhibit different ambiguity structures and stronger dependence on resource policies. The simulator blueprint assumes queue observability and stable activity taxonomies; in settings with concept drift, task renaming, or ad-hoc activities, both the predictor and the simulator would need incremental updates and drift-aware evaluation.

Construct validity. Our primary metrics focus on next-activity accuracy and confusion analysis, complemented by macro F1 to reflect tail classes. These are standard in PPM, yet they do not fully capture operational value. For example, a model that improves Top-1 by reassigning probability mass among frequent activities may have negligible effect on throughput time if the resource bottleneck remains unchanged. This motivates the proposed simulator metrics—global next-event accuracy, per-resource next-task precision, and workload MAPE—to better align evaluation with operational objectives. Finally, we fixed Top-3 as the selection criterion for early stopping; alternative criteria such as calibration error or cost-sensitive risk could yield different checkpoints with different deployment trade-offs.

# 8 Conclusion

We provided a reproducible, leakage-safe protocol and a compact LSTM baseline for next-activity prediction on three widely used event logs, together with a modular blueprint for resource-centric agents implemented via per-resource policies and discrete-event simulation. The baseline delivers strong Top-3 accuracy and competitive Top-1 across chronological test splits, while triangulating confusion patterns highlights ambiguities consistent with unmodeled resource contention. We documented pitfalls and a practical checklist spanning lifecycle handling, imbalance-aware back-offs, safe indexing, and artifact logging. Next steps include releasing the full simulator with standardized metrics and ablations, and integrating richer queue features and priority signals to better capture op-

erational dynamics. We hope these artifacts help the community build trustworthy, resource-aware PPM experiments.

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

# Supplementary Material

## A   Implementation details

Data loading. The discovery utility scans input directories for files with the extensions `.xes` or `.xes.gz`, preferring a local input folder. XES logs are parsed with PM4Py [1] into a tidy DataFrame with columns case_id, activity, lifecycle, timestamp, resource, with UTC timestamps and sorted by time and case.

Prefix construction. For each case, we sort events by timestamp, filter to lifecycle "complete", and derive per-step features: inter-event delta (seconds), since case start (seconds), hour of day in [0,1], weekday in [0,1], and a working-hours flag (Mon–Fri, 08–17). We emit prefixes of length $k \in [1, \min(10, T-1)]$ with target at position $k$. To avoid positional indexing bugs, timestamps are converted to numpy arrays prior to computing deltas.

Normalization. We compute mean and standard deviation for delta and since-start on training prefixes only (after chronological split), then apply the same transformation to validation and test prefixes.

Model and training. Activity IDs are embedded into 64 dimensions and concatenated with the five continuous features, then fed to a single-layer LSTM with hidden size 128 and dropout 0.2. The final hidden state goes to a linear classifier over the activity vocabulary. We train with cross-entropy loss and Adam using learning rate 1e-3, betas 0.9 and 0.999, epsilon 1e-8, batch size 128, and select the best epoch by validation Top-3 accuracy. Seeds are fixed at 42. Unless otherwise noted, there is no weight decay, no label smoothing, and no gradient clipping. We use token padding with masking so that loss is computed only on valid time steps.

Artifacts. We export per-dataset loss curves and test confusion matrices as PNGs. In the main text we keep the confusion matrices and relocate loss curves to the appendix to prioritize information density; additional artifact dumps are released with the code for deeper offline analysis.

## B Resource-centric simulator blueprint

State and events. The DES maintains (i) a global clock, (ii) per-resource busy/idle status and residual service times, and (iii) per-activity queues with counts and oldest waiting time. When a completion event occurs, the corresponding resource becomes idle and immediately selects the next activity.

Policies. Each resource $r$ has a multinomial logistic policy $\pi_r(a \mid x)$ over activities $a$ with features $x$ including previous activity executed by $r$, time-of-day bins, per-activity queue counts, and per-activity oldest waiting time. If samples for a class are below a threshold, we back off to a global policy $\pi_{\text{global}}$ fit on all resources.

Durations. Each activity $a$ has a log-normal distribution for service time with parameters fit from training "start"/"complete" pairs when available; otherwise, we use the sample median from "complete" inter-event deltas as a fallback for instantaneous transitions.

Dispatch and ablations. Given an activity choice, the resource dispatches the oldest waiting case in the selected activity queue (FIFO within activity). The FIFO ablation replaces $\pi_r$ by selecting the activity with the oldest waiting job across all queues, removing learning from the decision rule.

Metrics. We propose reporting: (1) global next-event accuracy comparing predicted next completion against the replayed next completion; (2) per-resource next-task precision; (3) workload MAPE comparing per-resource busy time profiles over evaluation horizons; and (4) standard PPM metrics (Top-k next-activity, remaining time MAE, suffix similarity) for completeness.

## C Consolidated learning curves

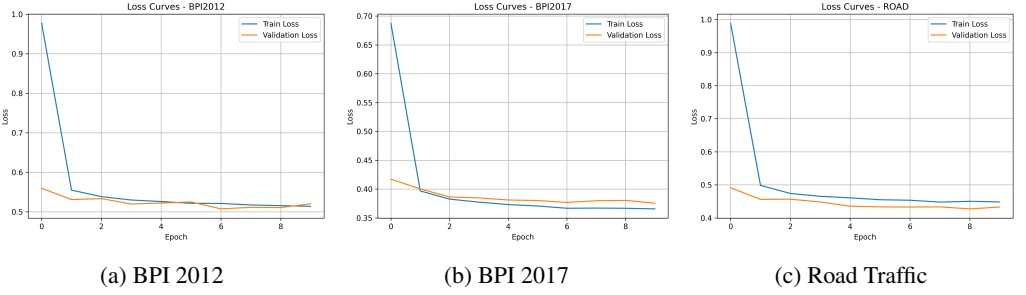

(a) BPI 2012        (b) BPI 2017        (c) Road Traffic

Figure 2: Training (blue) and validation (orange) loss curves for the LSTM baseline. Loss drops sharply in early epochs and then plateaus. The larger train-validation gap in BPI 2012 signals overfitting relative to BPI 2017 and Road Traffic.

## Agents4Science AI Involvement Checklist

1. **Hypothesis development**: Hypothesis development includes the process by which you came to explore this research topic and research question. This can involve the background research performed by either researchers or by AI. This can also involve whether the idea was proposed by researchers or by AI.

   Answer: **[C]**

   Explanation: Explanation: A postdoctoral researcher in BPM proposed the initial idea and provided a short JSON note with a sketch abstract, minimal experiment outline, and key limitations. A customized AI Scientist v2 (tuned for BPM/PPM) then expanded the problem framing, surveyed related work, refined the hypotheses, and generated alternative angles and ablations. Human input focused on scoping and feasibility; the AI did the majority of hypothesis refinement and articulation.

2. **Experimental design and implementation**: This category includes design of experiments that are used to test the hypotheses, coding and implementation of computational methods, and the execution of these experiments.

   Answer: **[D]**

   Explanation: The AI agent system produced the detailed experimental plan (data analysis, splits, features, baselines/ablations, metrics, and runtime constraints) and drafted implementation scaffolds consistent with our BPM/PPM customization prompts. The AI contributed all of the design specifics and executable structure and implemented end-to-end fully autonomous pipeline.

3. **Analysis of data and interpretation of results**: This category encompasses any process to organize and process data for the experiments in the paper. It also includes interpretations of the results of the study.

   Answer: **[D]**

   Explanation: The AI agent system generated data analyses (tables, confusion-matrix reads, error patterns, and suggested ablations) and implemented interpretation text. AI reviewed for domain correctness (e.g., concurrency/resource nuances), pruned wrong statements, and ensured that claims matched observed metrics and logs. Thus, AI carried the bulk of analysis drafting.

4. **Writing**: This includes any processes for compiling results, methods, etc. into the final paper form. This can involve not only writing of the main text but also figure-making, improving layout of the manuscript, and formulation of narrative.

   Answer: **[D]**

   Explanation: From outline to full manuscript (sections, figures/captions text, and references), drafting was done by the our AI agent system (customized version of AI Scientist v2 by Sakana AI). Final polishing (clarity, tone, formatting, and minor rewrites) used Chat-GPT as a reviewer/editor under human supervision. Humans provided high-level guidance and performed final compliance checks (style, anonymization) but did not author substantial portions of the prose nor change any claims made in the paper.

5. **Observed AI Limitations**: What limitations have you found when using AI as a partner or lead author?

   Description: The AI agent was able to act as a fully autonomous partner for baseline construction, ablations, and reproducible experimental analysis. We deliberately avoided human intervention during experiment execution and data analysis to test its autonomy. While it reliably handled standard tasks and produced consistent pipelines, it struggled to generate novel or complex experimental ideas beyond the templates it had been given. In practice, we found it best suited as a dependable assistant for systematic evaluation rather than as an originator of fundamentally new methodological contributions.

