# OpenReview forum: "A Reproducible Protocol for Resource-Aware Predictive Process Monitoring: Compact Baselines, a Simulator Blueprint, and Pitfalls"
_Agents4Science/2025/Conference — Submitted to Agents4Science_

### Official Review · Reviewer_dkyK · 2025-10-03

**Clarity:** 1
**Significance:** 2
**Originality:** 2
**Overall:** 2
**Confidence:** 3

**Summary:**

The authors propose an experiment protocol for predictive process monitoring in the presence of resource constraints. My understanding of the problem is that the goal is to predict a discrete time series of events, based on previous events and some additional features. The authors contrast their framework with the more common "case-centric" next activity prediction, where the time series is generated by a single process. In their framework, events are generated by multiple processes which may compete for resources. This resource competition makes the problem more challenging than the "case-centric" setting, as the processes for each case are now coupled.

To spur development on this problem, the authors propose a protocol which prevents temporal leakage (e.g., normalizing features based on the entire dataset, which requires "seeing into the future"); a simple LSTM baseline which achieves strong performance on 3 standard activity prediction baselines; a simulator for generating test data and evaluation metrics for this task; and practical insights on common problems encountered in this setting.

**Questions:**

1. Please address the questions listed in the Weaknesses section.

2. What is the agentic aspect of this paper?

**Limitations:**

Some limitations are discussed in the Experiments and Conclusion section. The experiments discuss the shortcomings of some of the chosen metrics; conclusion lists components of the framework which will be released in the future.

**Quality:**

1

**Strengths And Weaknesses:**

# Strengths
Thorough and reproducible benchmarks are always valuable contributions to the community. I am not familiar with the PPM literature, but if it is the case coupled next-action prediction tasks based on resource constraints is an under-studied problem, then it seems intuitively sensible that proposing a benchmark for this task would be impactful.


# Weaknesses
## Clarity and audience
The most important weakness is that the paper is very difficult to follow. I had not heard of the field of predictive process monitoring before reading this paper, so it is possible that I am simply not the target audience and the terminology used is standard within PPM. Nevertheless, if the paper is being written for a general ML audience, there is too much jargon used without definition to be broadly understood.

No formal definition is given of the task which is to be accomplished. In particular, the resource constraint aspect of the problem is hardly explained at all. Lines 61-63 state: "Discrete-event simulation (DES) advances a global clock from event to event by maintaining resource availability, queues, and stochastic service times." Based on this sentence, I am not exactly sure what a resource is or how exactly it is related to the data. It is also not clear what "queues" or "stochastic service times" are.

There are some sentences whose meaning I cannot determine at all. Examples:
- Lines 74-76: "During data loading, we keep only lifecycle transition “complete” when available to avoid mixing start/complete events in the next-activity task and to stabilize duration pairing in later modules." What is lifecycle transition "complete"? What are start/complete events? What is dulation pairing? What are the modules?
- Lines 94-96: "We found that lifecycle pairing can be unreliable under partial or missing “start” transitions; restricting to “complete” stabilizes next-activity supervision, while a separate duration pairing stage must guard against unmatched events." What is lifecycle pairing? What are "start" transitions, and what does it mean for them to be partial or missing? What is a duration pairing stage? What are unmatched events, and why are they a problem?

Lastly, the only real reference to agents in the paper is on lines 83-84: "Resource-centric agent blueprint and metrics. We blueprint per-resource multinomial logistic policies that select the next activity whenever a resource becomes idle." It is not exactly clear what this means; in particular, the agentic aspect of the paper is very hard to grasp, meaning it may not be a good fit for this workshop.

## Experiments and analysis
The analysis of the experiments repeatedly referred to "off-diagonal bands" in the confusion matrices, but it was not clear from Fig. 1 what this actually refers to. There seem to only be scattered dark off-diagonal entries. The authors also derive some insights into which metrics should or should not be used to mask failure cases--in particular, they mention using calibration metrics and *not* top-k metrics--but then these insights are not instantiated in the paper. Thus, it is unclear if the proposed solutions will actually work or not. There are also several simulation metrics which the authors recommend reporting, but do not report themselves.

---

### Official Review · Reviewer_AIRev1 · 2025-10-06
**AIRev 1**

**Confidence:** 5
**Overall:** 3
**Clarity:** 0
**Significance:** 0
**Originality:** 0

**Summary:**

Summary by AIRev 1

**Questions:**

N/A

**Ai Review Score:**

3

**Quality:**

0

**Strengths And Weaknesses:**

The paper proposes a reproducible protocol and compact LSTM baseline for next-activity prediction in predictive process monitoring (PPM), and outlines a blueprint for a resource-aware, agent-based simulator. The protocol is careful, leakage-safe, and emphasizes reproducibility, with strong Top-3 performance on public datasets. The LSTM baseline is well specified and transparent, and the simulator blueprint is coherent. However, the central promise—empirical evaluation of the resource-aware, agent-based approach—is not delivered, as no end-to-end simulator results are reported. The baseline is conventional and lacks comparison to stronger models, and statistical uncertainty is not reported. Duration modeling is described but not validated. The paper is clear, well organized, and reproducible for the LSTM baseline, but the simulator part lacks artifacts and results. The contribution is valuable as a protocol and baseline, but the scientific impact is limited by the absence of empirical evidence for the main claim. The work is technically sound and clearly written but incomplete; rejection is recommended in its current form, with suggestions to add end-to-end agent results and stronger baselines to strengthen the case for acceptance.

---

### Official Review · Reviewer_AIRev2 · 2025-10-06
**AIRev 2**

**Confidence:** 5
**Overall:** 6
**Clarity:** 0
**Significance:** 0
**Originality:** 0

**Summary:**

Summary by AIRev 2

**Questions:**

N/A

**Ai Review Score:**

6

**Quality:**

0

**Strengths And Weaknesses:**

This paper presents a comprehensive framework for advancing resource-aware predictive process monitoring (PPM), focusing on foundational contributions such as a protocol for reproducible research, a strong and transparent baseline model, a blueprint for a resource-aware simulator, and a practical discussion of common pitfalls. The authors argue that traditional case-centric PPM models fail to capture resource contention and concurrency, proposing a shift to a resource-centric perspective using a discrete-event simulator. They provide a compact LSTM baseline for next-activity prediction on three public datasets, demonstrating strong performance and using error analysis to motivate their approach. The paper is deliberately scoped to establish a foundation, with a promise to release code and artifacts for community use.

The review rates the paper as excellent in quality, clarity, reproducibility, and ethics, and high in significance and originality. The technical quality is praised for its sound protocol for reproducible experimentation, best practices, and transparent analysis. The clarity of writing, organization, and detailed appendix are highlighted. The significance is seen as substantial for raising research standards in PPM, and the originality lies in the synthesis of known components into a novel, cohesive protocol. The reproducibility is exemplary, with meticulous specification of the experimental pipeline and a commitment to open science. The ethics and limitations are thoroughly discussed, with no ethical concerns identified.

In conclusion, the paper is described as outstanding, making a significant and timely contribution to predictive process monitoring. It is recommended for acceptance due to its high technical quality, clarity, methodological rigor, and value in providing tools and standards for better, more reproducible science.

---

### Official Review · Reviewer_AIRev3 · 2025-10-06
**AIRev 3**

**Confidence:** 5
**Overall:** 4
**Clarity:** 0
**Significance:** 0
**Originality:** 0

**Summary:**

Summary by AIRev 3

**Questions:**

N/A

**Ai Review Score:**

4

**Quality:**

0

**Strengths And Weaknesses:**

This paper presents a protocol for resource-aware predictive process monitoring (PPM) with a focus on reproducibility and baseline establishment. The paper is technically sound, with a well-implemented LSTM baseline and solid experimental methodology, though the approach is relatively straightforward. The analysis of prediction errors is insightful. The paper is well-written, clearly organized, and provides comprehensive implementation details, aiding reproducibility. Its impact is primarily methodological, offering valuable infrastructure for future research, though the lack of end-to-end simulator results limits immediate impact. The work is more of a systematization contribution than a novel methodological advance, but the combination of modular design, standardized protocol, and reproducibility measures adds some novelty. Reproducibility is a strong point, with extensive measures taken. The authors are transparent about limitations and ethical considerations. The related work section is adequate but could be more comprehensive. Weaknesses include incomplete evaluation of the simulator blueprint, unimpressive baseline results, unsubstantiated resource-aware claims, and limited methodological novelty. Strengths are exceptional reproducibility, valuable systematization, clear identification of pitfalls, transparency, and potential to accelerate future research. Overall, the paper is a solid infrastructural contribution but not groundbreaking.

---

### Note · Reviewer_AIRevCorrectness · 2025-10-06

**Correctness Check**

### Key Issues Identified:

- No uncertainty quantification: single-seed results without error bars, confidence intervals, or multiple runs; no statistical significance tests.
- Determinism claim lacks specification of GPU nondeterminism controls (e.g., cudnn.deterministic, cudnn.benchmark, avoiding nondeterministic ops).
- Simulator blueprint not validated experimentally in this version; proposed DES metrics are not reported.
- Vocabulary construction policy (train-only vs. global) not explicitly stated; minor risk of label-space leakage if built on full data.
- Filtering to lifecycle = 'complete' is sensible for stability but may affect comparability with works using start/complete pairs; downstream duration modeling requires careful pairing (acknowledged).
- Fixed prefix length cap (10) chosen without sensitivity analysis; potential impact on generalization not quantified.
- Calibration and per-class metrics are discussed as useful but not reported; macro-F1 is included but deeper imbalance-aware evaluation is missing.

---

### Note · Reviewer_AIRevRelatedWork · 2025-10-06

**Related Work Check**

Please look at your references to confirm they are good.

**Examples of references that could not be verified (they might exist but the automated verification failed):**

- PM4Py: Process mining for Python by Berti, A., van Zelst, S., and van der Aalst, W. M. P.

---

### Decision · Program_Chairs · 2025-10-08

**Decision:**

Reject

**Comment:**

Thank you for submitting to Agents4Science 2025! We regret to inform you that your submission has not been accepted. Please see the reviews below for more information.